# Pulmonary and Gastrointestinal Parasitic Infections in Small Ruminant Autochthonous Breeds from Centre Region of Portugal—A Cross Sectional Study

**DOI:** 10.3390/ani14081241

**Published:** 2024-04-21

**Authors:** Maria Aires Pereira, Maria João Vila-Viçosa, Catarina Coelho, Carla Santos, Fernando Esteves, Rita Cruz, Liliana Gomes, Diogo Henriques, Helena Vala, Carmen Nóbrega, Ana Cristina Mega, Carolina de Melo, Madalena Malva, Joana Braguez, Teresa Letra Mateus

**Affiliations:** 1Instituto Politécnico de Viseu, Escola Superior Agrária de Viseu, Campus Politécnico, 3504-510 Viseu, Portugal; ccoelho@esav.ipv.pt (C.C.); casarede@esav.ipv.pt (C.S.); festeves@esav.ipv.pt (F.E.); rcruz@esav.ipv.pt (R.C.); lilianamonica321@gmail.com (L.G.); diogo_henriques07@hotmail.com (D.H.); hvala@esav.ipv.pt (H.V.); cnobrega@esav.ipv.pt (C.N.); amega@esav.ipv.pt (A.C.M.); 2Global Health and Tropical Medicine, GHTM, Associate Laboratory in Translation and Innovation Towards Global Health, LA-REAL, Instituto de Higiene e Medicina Tropical, IHMT, Universidade NOVA de Lisboa, UNL, Rua da Junqueira 100, 1349-008 Lisboa, Portugal; 3CERNAS-IPV Research Centre, Instituto Politécnico de Viseu, Campus Politécnico, Repeses, 3504-510 Viseu, Portugal; 4Laboratório de Parasitologia Victor Caeiro, Departamento de Medicina Veterinária, Universidade de Évora—Pólo da Mitra, Apartado 94, 7002-554 Évora, Portugal; mjoaovv@uevora.pt; 5MED, Instituto Mediterrâneo para a Agricultura, Ambiente e Desenvolvimento, Universidade de Évora—Pólo da Mitra, Apartado 94, 7006-554 Évora, Portugal; 6Veterinary and Animal Research Centre (CECAV), UTAD, Associate Laboratory for Animal and Veterinary Sciences (AL4AnimalS) Quinta de Prados, 5000-801 Vila Real, Portugal; tlmateus@esa.ipvc.pt; 7EpiUnit—Instituto de Saúde Pública da Universidade do Porto, Laboratory for Integrative and Translational Research in Population Health (ITR), Rua das Taipas, nº 135, 4050-091 Porto, Portugal; 8Centre for the Research and Technology of Agro-Environmental and Biological Sciences (CITAB), University of Trás-os-Montes e Alto Douro, 5001-801 Vila Real, Portugal; 9Instituto Politécnico de Bragança, Alameda de Santa Apolónia 253, 5300-252 Bragança, Portugal; carochamelo@gmail.com; 10Instituto Politécnico de Viseu, Escola Superior de Tecnologia e Gestão de Viseu, Campus Politécnico, 3504-510 Viseu, Portugal; madalena.malva07@gmail.com (M.M.); jbraguez@estgv.ipv.pt (J.B.); 11CIAC—Centro de Investigação em Artes e Comunicação Universidade do Algarve, Campus Gambelas, Sala 0.28, Edifício 1, FCHS, 8005-139 Faro, Portugal; 12CEIS20—Centro de Estudos Interdisciplinares, Rua Filipe Simões nº 33, 3000-186 Coimbra, Portugal; 13CISAS—Center for Research and Development in Agrifood Systems and Sustainability, Escola Superior Agrária, Instituto Politécnico de Viana do Castelo, Rua Escola Industrial e Comercial de Nun’Àlvares, 4900-347 Viana do Castelo, Portugal

**Keywords:** sheep, goat, lungworm infection, Protostrongylidae, *Muellerius capillaris*, *Dictyocaulus filaria*, risk factors

## Abstract

**Simple Summary:**

The production of small ruminant autochthonous breeds in the Centre region of Portugal is practiced under grazing, exposing animals to parasitic infections. The main objective of this study was to estimate the prevalence of lungworm parasitic infection and identify risk factors to define appropriate control measures. Fecal samples of 203 goats and 208 sheep from 30 herds, located in three districts in the Centre region of Portugal, were collected and analyzed. The overall prevalence of lungworm infection was 57.7%, significantly higher in goats (95.6%) than in sheep (20.7%). The risk of lungworm infection was 29.7, 7.4, or 8.7 times higher for sheep dewormed with albendazole, mebendazole plus closantel, or ivermectin plus clorsulon, respectively, than for those dewormed with eprinomectin. Additionally, the presence of gastrointestinal parasites was investigated in 307 fecal samples and an overall prevalence of infection of 86.3% was observed, also significantly higher in goats (93.2%) than in sheep (79.9%). Considering the high prevalence and the burden of lungworm parasitic infection, it is urgent to determine its economic impact and the repercussions in animal health in the Centre region of Portugal to establish appropriate therapeutic guidelines.

**Abstract:**

The production of small ruminant autochthonous breeds in the Centre region of Portugal is practiced in a semi-extensive husbandry system, exposing animals to parasitic infections. The main objective of this study was to estimate the prevalence of lungworm infection and identify risk factors. Fecal samples of 203 goats and 208 sheep from 30 herds were collected *per rectum* and subjected to the modified Baermann test. The overall prevalence of infection was 57.7%, significantly higher in goats (95.6%) than in sheep (20.7%) (*p* < 0.001). According to the binary logistic regression model, sheep dewormed with albendazole, mebendazole plus closantel, or ivermectin plus clorsulon presented a risk of Protostrongylidae infection 29.702, 7.426, or 8.720 times higher, respectively, than those dewormed with eprinomectin. Additionally, the presence of gastrointestinal parasites was investigated in 307 fecal samples using Mini-FLOTAC^®^. The overall prevalence of infection was 86.3%, also significantly higher in goats (93.2%) than in sheep (79.9%) (*p* < 0.001). Strongyle-type eggs were the most frequently identified, both in sheep (69.8%) and goats (87.8%), followed by *Eimeria* oocysts (40.3% in sheep and 68.9% in goats). Considering the high prevalence and the burden of lungworm parasitic infection, it is urgent to determine its economic impact and the repercussions in animal health in the Centre region of Portugal to establish appropriate therapeutic guidelines.

## 1. Introduction 

The production of small ruminants in the NUTS II (Nomenclature of Territorial Units for Statistics) [1] Centre region of Portugal, essentially in semi-extensive husbandry system, has a positive impact on the local economy, playing an essential role in maintaining rural communities and preserving the ecosystems by controlling the biomass and preventing fires [2]. 

In Portugal, there are 16 sheep and six goat autochthonous breeds, which constitute a unique genetic heritage [3], but most of them are in danger of extinction. Portuguese small ruminant autochthonous breeds are rustic and adapted to different and adverse soil and climate conditions [4], taking advantage of forage resources that otherwise would not be used [5]. 

Most autochthonous breeds have dual aptitude, and their meat and milk are of high nutritional and organoleptic quality, being the main source of income for rural populations in the interior and mountainous regions [4]. However, the low fertility and prolificity of these breeds have led to the introduction of males of exotic breeds to increase reproductive efficiency and productivity of herds [3]. 

Serra da Estrela is an autochthonous sheep breed reared for a dairy purpose. The milk is used to produce “Serra da Estrela” cheese, a “Protected Designation of Origin” (PDO) product, highly valued both in the country and internationally, and the lambs sold at one month of age are designated “Borrego Serra da Estrela” (PDO). The autochthonous goats breed Serrana have four ecotypes, the Transmontano, Jarmelista, da Serra, and Ribatejano, which are exploited in an extensive system with dual purpose [4].

Grazing ruminants are inevitably exposed to helminth parasites, which negatively impact on feed intake, growth, mortality rates, carcass weight and quality, wool growth, fertility, and milk yield [6,7]. The cost of helminth infections in the European ruminant livestock industry was estimated in EUR 2.1 billion [7]. The conservation, sustainable use and promotion of animal genetic resources may reduce these costs, since autochthonous small ruminant breeds seem to be more resistant and resilient to parasitic infections [8,9]. 

The main helminth parasites described in small ruminants infect the respiratory and digestive systems. Lungworm species affecting small ruminants are *Dictyocaulus filaria* and the Protostrongylidae, *Muellerius capillaris*, *Protostrongylus rufescens*, *Cystocaulus ocreatus,* and *Neostrongylus linearis*. *D. filaria* has a direct life cycle in which the first-stage larvae (L_1_) are shed via feces, developing into L_3_ in the environment. L_3_ are ingested by ruminants while grazing and migrate to the respiratory system where adult worms mature within the lumen of bronchi and bronchioles [10]. Species of the Protostrongylidae family have an indirect life cycle, requiring the intervention of an intermediate host (terrestrial gastropod) for the development of L_1_ into L_3_ infective stage. Small ruminants became infected via ingestion of slugs and snails harboring L_3_. In both cases, adult females lay eggs in the terminal bronchioles and alveoli and L_1_ are coughed up and swallowed with the respiratory secretion, being excreted in feces [11].

*D. filaria* infection frequently results in clinical respiratory signs, and production losses have been documented [10]. However, Protostrongylidae infection is usually subclinical, except for occasional coughing. Economic losses, resulting from reduced growth and carcass weights; increased susceptibility to other diseases; and rejection at slaughter were documented in some regions [12,13,14] but not consistently [15].

Antemortem diagnosis of lungworm infection is mainly carried out using the Baermann technique, which consists of recovering L_1_ from the feces and identify larvae species via microscopy (larvoscopy) based on its morphological characteristics [16]. Despite being non-invasive, simple, and inexpensive, the Baermann technique has suboptimal sensitivity. Indeed, sensitivity of the Baermann technique did not exceed 80% for *M. capillaris* in sheep using pooled samples [17,18].

Sheep and goats harbor and share a variety of gastrointestinal parasitic species [19,20]. Although most gastrointestinal parasitic diseases are caused by nematodes, trematodes and cestodes also contribute to economic losses in small ruminant production [21]. Despite the regular deworming of the herds, gastrointestinal parasites are still a major problem in small ruminant production, and the emergency of anthelminthic resistance is an increasing concern [2,22,23].

To our knowledge, studies on lungworm and gastrointestinal parasitic infections in small ruminants in Portugal are scarce. Understanding the risk factors is essential to design effective antiparasitic treatment and/or prevention programs [24]. Thus, the main objective of this study was to estimate the prevalence and the burden of lungworm infection and identify risk factors to define appropriate control measures. Additionally, we intended to determine the prevalence and burden of gastrointestinal parasitic infection and contribute to increasing our knowledge about parasitic infections in autochthonous small ruminants breeds in NUTS II Centre region of Portugal.

## 2. Materials and Methods

### 2.1. Study Area

This study was carried out in the NUTS II Centre region of Portugal, specifically in the districts of Viseu, Guarda, and Coimbra. The Centre region occupies an area of 23,666 km^2^ [25] and the districts of Viseu, Guarda, and Coimbra extend through an area of 5007 km^2^, 5518 km^2^, and 3947 km^2^, respectively. The Centre region is characterized by the presence of the largest mountain range of the country, which culminates in Serra da Estrela (altitude of 1991 m) [25]. Considering the Köppen–Geiger classification (1971–2000), the climate in the Centre region is temperate (type C) with subtype Cs (temperate climate with dry summer). The predominant variety is Csb, corresponding to a temperate climate with dry and mild summer, although in some areas the variety is Csa, corresponding to temperate climate with hot and dry summers [26] (Figure 1). The small ruminant population in the Centre region comprises 549,700 animals, distributed across 24,704 herds [27]. In this region, the production system is composed of small size herds, and animals are reared in a semi-extensive husbandry system, consisting of grazing during the day and housing during the night [4]. Transhumance, the seasonal movement of herds and shepherds to locations that offer better conditions during part of the year, is an ancestral tradition in the Centre region of Portugal, involving Serra da Estrela, Gardunha, and Montemuro [28]. The animals’ diet consists of natural pastures, shrub vegetation, cereal stubble, and sometimes sown pastures. When there is little availability of pasture or the dietary needs are more demanding, the animals are supplemented with forage, cereals, and/or commercial concentrates [4].

### 2.2. Herd and Animal Sampling

Samples were collected on the day the animals were dewormed and subjected to the official sanitary program, which included annual blue tongue vaccination and blood collection for Brucellosis testing. Herd selection was carried out for convenience, being included in the study small ruminant herds visited by the official veterinary brigade between March and June 2023 that had not been dewormed in the previous six months. 

In this region, sheep herds are of medium size. To sample representative herds of the region, only those with 25 or more sheep were included in the study. The size of sampled herds ranged between 25 and 242 heads (mean 88 heads). A total of 15 sheep aged over six months old were randomly sampled *per* herd.

However, since in this region goat herds are generally small, the totality of the animals from each goat herd aged over six months old were sampled. The size of the 16 sampled herds varied between 6 and 22 heads (mean of 13 heads).

Surveyed animals were from Portuguese autochthonous purebreds or crossbreds: Serra da Estrela sheep breed, Serrana ecotype Jarmelista goat breed, and Serrana ecotype Transmontano goat crossbreed (Figure 2).

### 2.3. Questionnaire

A questionnaire comprising data about the animal signalment (species, breed, age, and aptitude); husbandry management systems, including pasture sharing with other small ruminant herds; and parasitological information (testing, frequency of deworming, and anthelmintic compound used in the last treatment) was completed on the day of sampling.

### 2.4. Fecal Samples

Fecal samples of 208 sheep and 203 goats were analyzed to assess the prevalence and the burden of lungworm infection using the modified Baermann technique. 

Whenever lungworm analysis did not consume the entire sample, and complying with the principles of Replacement, Reduction, and Refinement (3Rs), the remaining feces were used to investigate gastrointestinal parasites, assessing its prevalence and the burden of infection, and identify *Eimeria* species, contributing to increase the knowledge about parasitic diseases in small ruminant autochthonous breeds from Portugal. 

The amount of feces collected from 148 sheep and 159 goats was sufficient to evaluate the prevalence and the burden of gastrointestinal parasitic infection using Mini-FLOTAC^®^ (University of Napoles Federico II, Naples, Italy). Additionally, *Eimeria* species present in fecal samples from 40 sheep and 52 goats were identified. 

Fecal samples were collected directly from each animal *per rectum* using plastic gloves and stored in plastic trays in a cooling container during sample collection and transport to the laboratory. Samples were processed on the same day of collection using the Baermann technique, and the remaining feces were analyzed via Mini-FLOTAC^®^ and *Eimeria* species were identified.

### 2.5. Modified Baermann Technique 

Each individual 5 g fecal sample was formed into a small pat and suspended in a gauze within a plastic cup. The cup was filled with water to just cover the fecal sample and incubated at room temperature for 12 h. After the incubation period, the water was carefully removed, keeping 5–10 mL of sediment which was homogenized and transferred to a Petri dish for observation under a magnifying glass (×40). The number of larvae per gram of feces (LPG) was estimated using a McMaster chamber under an optical microscope, and L_1_ were identified according to the characteristics of the posterior section [15]. Five larvae were collected from positive samples, observed between slide and coverslip after immobilization by adding a drop of 1% iodine solution, and were photographed. 

### 2.6. Eggs/Oocysts Per Gram of Feces

Concerning gastrointestinal coprology, the quantitative technique used to analyze samples was Mini-FLOTAC^®^ supported by a Fill-FLOTAC^®^ Kit (University of Napoles Federico II, Naples, Italy), according to Cringoli et al. (2017) [29]. Whenever possible (i.e., when enough sample quantity was available) two flotation solutions were used—saturated sodium chloride (NaCl, specific gravity = 1.200) and zinc sulphate (ZnSO_4_, specific gravity = 1.350)—according to the instructions reported in the original description by Cringoli et al. (2017) [29]. For each sample, two aliquots of 5 g of feces were added with 45 mL of each solution and homogenized in the Fill-FLOTAC, as described. The two flotation chambers of the Mini-FLOTAC disc were filled and after 10 min the top part of the disc was rotated and then ready to be read on a light microscope. Magnifications of 100× and 400× were used to identify parasitic forms (helminths eggs and protozoan oocysts). The parasitic forms were counted and multiplied by 5 to obtain the number of eggs/oocysts per gram (EPG/OPG) of feces.

### 2.7. Eimeria Species Identification

After being isolated from the feces, oocysts were allowed to sporulate, as previously reported [24]. The oocyst suspension was transferred to Petri dishes covered with small glass panels. Oocysts that bonded to the glass panels were washed off with water and were added to 2% (*w*/*v*) potassium dichromate solution to allow complete sporulation at room temperature and with regular aeration for 2–5 days. After concentration, simple flotation using saturated saline solution (NaCl, specific gravity = 1.200) was performed and the oocysts were identified under a microscope based on its shape and size, presence or absence of micropyle and polar cap, color and thickness of oocyst wall, presence or absence of Sieda body at the more pointed end of sporocysts, and presence or absence of oocystic and sporocystic residual body [24]. 

### 2.8. Data Processing and Statistical Analysis

The background data collected during the official veterinary brigade visit and the results of parasitological analyses were downloaded in a database (Microsoft Excel 2016^®^; Microsoft Corp., Redmond, WA, USA). Statistical analysis was performed using IBM SPSS v.28.0.0.0 (IBM Corp., Armonk, NY, USA, 2020) applying descriptive and inferential statistical analysis. Prevalence and confidence intervals (CI) of lungworm and gastrointestinal parasitic infections were calculated. The mean and standard deviation (SD) of LPG and EPG/OPG were also determined. Lungworm infection burden was classified as light (1–150 LPG), moderate (151–500 LPG), or heavy (>500 LPG) and the distribution of LPG was studied for both species (sheep and goats), and the results are presented as box plots. The non-parametric Mann–Whitney test for independent samples was used to compare the distribution of numerical variables between animal species (sheep vs. goats) at the 5% level of significance. Univariable analysis via Chi-square testing was performed to evaluate the association between Protostrongylidae infection (dependent variable) and the qualitative independent variables (district, pasture sharing, production purpose, sex, breed, age, parasitological testing, deworming frequency, and dewormer), for both species (sheep and goats) at a level of significance of 5% (*p* < 0.05). Two binary logistic regression models were constructed (sheep and goats) to identify risk factors associated to Protostrongylidae infection. Statistically significant (*p* < 0.05) variables identified in the univariate analysis were included in the analysis and the best models were defined using the forward stepwise technique. Odds ratios (OR) were determined at a 5% level of significance. 

### 2.9. Ethics

Fecal samples were collected with the permission of farm owners and according to good veterinary practices and animal welfare standards. Experimental procedures were performed according to the European Directive 2010/63/EU on the protection of animals used for scientific purposes and was approved by the Órgão para o Bem-Estar Animal (ORBEA) of Escola Superior Agrária de Viseu (ESAV) with the reference 01/ORBEA/2023.

## 3. Results

### 3.1. Characterization of Sampled Animals and Herds

Background information collected allowed the characterization of the animals and herds.

A total of 411 animals from 30 herds were sampled, of which 208 (50.6%) were sheep and 203 (49.4%) were goat. All sheep belonged to the Serra da Estrela breed (208; 50.6%), while the goats belonged to the Serrano ecotype Transmontano crossbreed (118; 45.7%) or Serrano ecotype Jarmelista (15; 3.6%). Most animals were female (95.6%) and aged more than 12 months (94.9%). 

Most producers/farmers did not perform parasitological tests (96.7%) to monitor the prevalence and diversity of parasites. In most herds, animals were dewormed once a year (60.3%). Albendazole was the anthelmintic compound most frequently employed in goat herds (93.8%), while in sheep herds the combination of ivermectin plus clorsulon was the most used (42.9%) (Table 1).

### 3.2. Prevalence of Lungworm Infection and Infection Burden

Fecal samples of 411 small ruminants (208 sheep; 203 goats) from 30 herds (14 sheep; 16 goats) were analyzed using the modified Baermann technique and LPG was estimated. 

Lungworm infection was detected in 57.1% (8/14) of the sheep herds and in the totality (16/16) of the goat herds surveyed (Figure 3).

The overall prevalence of lungworm infection at animal level was 57.7%, significantly higher in goats (95.6%) than in sheep (20.7%) (*p* < 0.001). Goats were exclusively infected by species of the Protostrongylidae family, but sheep were infected by species of the Protostrongylidae family (17.3%), *D. filaria* (1.4%), or presented mixed infection (1.9%). Protostrongylidae were microscopically identified as *Muellerius capillaris* and *Cystocaulus ocreatus*. *M. capillaris* was identified in 42 out of the 43 positive sheep and in all the 194 positive goats (Table 2, Figure 4).

The overall burden of infection, measured as mean ± standard deviation of LPG, was 246.6 ± 413.8, significantly higher in goats (276.0 ± 442.4) than in sheep (92.1 ± 124.6) (*p* < 0.001). The minimum and maximum values of LPG were 0.2 and 646.0 in sheep and 1.5 and 3003.0 in goats, respectively (Figure 5).

The burden of infection was light in thirty-three (78.6%), moderate in eight (19.0%), and heavy in one (2.4%) of the lungworm-positive sheep. In goats, the burden of infection was considered light in 112 (55.7%), moderate in 50 (25.8%), and heavy in 32 (16.5%) animals.

### 3.3. Risk Factors Associated with Protostrongylidae Infection

A statistically significant association was observed between sheep Protostrongylidae infection and the variables district, pasture sharing, parasitological testing, and dewormer used in the last anthelminthic treatment (Appendix A). 

To evaluate the risk factors associated with sheep Protostrongylidae infection, a binary logistic regression was constructed. The final model presented an R^2^ Cox & Snell of 0.163 and R^2^ Nagelkerke of 0.255. The model sensitivity and specificity were 25.6% and 97.6%, respectively (global percentage of 82.7%). The final model included the variables pasture sharing and dewormer. According to the model, not sharing pasture with other small ruminant species was identified as a significant protection factor for Protostrongylidae infection (OR = 0.182; *p* = 0.001) compared with sharing pasture. Animals dewormed with albendazole presented a risk of infection 29.702 (*p* = 0.003) times higher than those dewormed with eprinomectin. Treatment with mebendazole plus closantel increased the risk of infection 7.426 (*p* < 0.001) times compared with eprinomectin. Animals previously treated with ivermectin plus clorsulon presented a risk of infection 8.720 (*p* < 0.001) times higher than those dewormed with eprinomectin (Table 3).

A statistically significant association was observed between goat Protostrongylidae infection and the variables production purpose, breed, deworming frequency, and dewormer compound (Appendix A). 

To evaluate the risk factors associated with goat Protostrongylidae infection, a binary logistic regression was constructed. The final model presented an R^2^ Cox & Snell of 0.113 and R^2^ Nagelkerke of 0.371. The model sensitivity and specificity were 100.0% and 0, respectively (global percentage of 95.6%). The final model included the variables production purpose and breed. According to the model, being reared for meat production was identified as a non-significant protection factor for Protostrongylidae infection (OR = 0.164; *p* = 0.150) compared with being reared for milk production. The Serrana ecotype Jarmelista breed presented a non-significant increased risk (OR = 9.333; *p* = 0.054) to be infected by Protostrongylidae compared with the Serrana ecotype Transmontano crossbreed (Table 4). 

### 3.4. Prevalence of Gastrointestinal Parasites and Infection Burden

Fecal samples of 307 small ruminants (148 sheep; 159 goats) from 28 herds (14 sheep; 14 goats) were analyzed using Mini-FLOTAC^®^ and EPG/OPG was estimated. 

Gastrointestinal parasites were identified in all herds surveyed. The overall prevalence of gastrointestinal parasitic infection at the animal level was 86.3% (CI: 0.821–0.898). Mixed infection (60.0%; CI: 0.540–0.658) was more frequent than single infection (40.0%; CI: 0.342–0.460). The prevalence of infection in goats (93.2%) was higher than in sheep (79.9%) (*p* < 0.001). In sheep, single gastrointestinal parasitic infection (52.0%) was more frequent than mixed infection (48.0%), whereas in goats mixed infection (71.0%) was more frequent than single infection (29.0%). 

All herds were considered positive for Strongyle-type eggs. Furthermore, Strongyle-type eggs were the most frequently identified at the animal level, both in sheep (69.8%) and goats (87.8%), followed by *Eimeria* oocysts (40.3% in sheep and 68.9% in goats) that were identified in most sheep herds (13/14) and in the totality of goat herds. Cestode infection was identified in 28.6% and 50.0% of the sheep and goat herds, respectively. The frequency of infection at the animal level was 3.8% in sheep and 8.1% in goats. *Trichuris* eggs were identified in two sheep herds and in one goat herd, specifically in five animals (four sheep and one goat). Eggs of *Skrjabinema*, a parasite belonging to the Oxyuridae family, were identified in two goat herds. Eggs of the Trematode *Dicrocoelium* were observed in one sheep (Table 5, Figure 6).

The burden of infection was higher for Strongyle-type eggs in both species. However, the mean burden of infection was significantly higher in goats (mea*n =* 518.0 EPG) than in sheep (mea*n =* 210.4 EPG) (*p* < 0.001). The burden of *Eimeria* oocysts was also higher in goats (mea*n =* 399.8 OPG) than in sheep (mea*n =* 99.6 OPG) (*p* < 0.001). Contrary, the burden of *Moniezia* eggs was higher in sheep (mea*n =* 105.8 EPG) samples than in goat samples (mea*n =* 75.8 EPG), but the differences were not statistically significant. *Trichuris*, *Skrjabinema,* and *Dicrocoelium* mean burdens of infection were low, ranging between 7.5 and 30.0 EPG (Table 6).

### 3.5. Identification of Eimeria Species

Oocyst suspension of feces recovered from 40 sheep (four herds) and 52 goats (four herds) were left to incubate with potassium dichromate solution to reach complete sporulation. Six *Eimeria* species were observed in sheep samples. Mixed infection (67.5%) was more frequent than single infection (32.5%). The most prevalent *Eimeria* species were *E. bakuensis* (67.5%) and *E. ovinoidalis* (45.0%). In goats, seven *Eimeria* species were identified and mixed infection (88.5%) was more frequent than single infection (11.5%). The most prevalent *Eimeria* species were *E. arloingi* (90.4%) and *E. ninakholyakimovae* (80.8%) (Table 7, Figure 7).

## 4. Discussion

Small ruminant infection by pulmonary nematodes is underdiagnosed, since lungworm testing is not included in the routine parasitological tests requested by veterinarians to clinical laboratories; when associated with the scarcity of scientific research, this creates a gap in the knowledge of pulmonary parasitic infections in Portugal, compromising its treatment and control. Thus, this study aimed to investigate the prevalence and burden of lungworm infection and identify associated risk factors. 

In the present study, the prevalence of lungworm infection reached 95.6% in goats and 20.7% in sheep. *M. capillaris* larvae were the most frequently observed, although *D. filaria* and *C. ocreatus* larvae have been sporadically observed in sheep samples. 

In line with our prevalence results, a parasitological study performed in goat herds located in the Gerês–Xurés Transboundary Biosphere Reserve (GXTBR), North region of Portugal revealed a Protostrongylidae prevalence of 100% in pooled fecal samples [30]. 

Comparing our results with those observed in Galiza, northwestern Spain, in our study, Protostrongylidae prevalence was higher and *D. filaria* prevalence was lower than those observed in Galiza (78.6% and 10.7% in goat and 11.6% and 10.6% in sheep, respectively) [31]. The burden of Protostrongylidae infection observed in goats (276.0 LPG) was in line with the results obtained by García-Dios et al. (2021) [31] (283.2 LPG). However, the burden of lungworm infection in sheep (92.1 LPG) was higher than that observed in Spain (11.9 LPG for Protostrongylidae and 8.5 LPG for *D. filaria*) [31]. The differences observed in lungworm prevalence, parasitic diversity, and burden of infection between the two regions may be attributed to pasture management and anthelmintic control carried out by producers.

Pasture management, namely sheep–goat mixed management systems seem to increase the risk of parasitic infection in sheep, and it has been suggested that goats are a source of contamination of pastures with parasites common to both species, namely lugworms [31]. In line with this, our model of binary logistic regression identified not sharing pastures as a protective factor for Protostrongylidae infection in sheep.

It has been suggested that a count of 150 LPG is indicative of parasitosis [32]. Surprisingly, in this study producers/farmers did not report clinical signs associated with lungworm infection, despite the high burden of infection observed. Although the low pathogenicity of lungworms and the predominantly asymptomatic nature of the infection did not predict significant direct economic losses, the truth is that the negative impact on animal health and productivity are difficult to ascertain, since lungworm infection often occurs in association with gastrointestinal parasitic infections and other co-morbidities [33].

*M. capillaris* was the most prevalent pulmonary parasite, as previously observed in Spain [34]. López et al. (2011) [34] suggested that the low pulmonary parasitic diversity may be related to generalized anthelmintic treatment used to control gastrointestinal nematodes. The maintenance and the high prevalence of *M. capillaris* in small ruminant populations may be related to the poor effectiveness of the anthelmintic compounds in these species. It was suggested that the larval stages of *M. capillaris* may be resistant to anthelmintic compounds, maturing after the destruction of adult parasites, which restart L1 excretion soon after treatment [35]. Pulmonary pathological changes induced by *M. capillaris*, characterized by the formation of inflammatory nodules, may protect parasites, preventing anthelmintic therapeutic concentrations in lung parenchyma [36].

Previous studies indicated that benzimidazoles and ivermectin have reduced efficacy against lungworm infection, especially against species that inhabit the lung parenchyma [35,37]. In line with this, our results revealed that sheep regularly dewormed with albendazole, mebendazole, or ivermectin had a risk of infection 29.7, 7.4, or 8.7 times higher than those treated with eprinomectin. Indeed, eprinomectin has shown reliable results in eliminating larval excretion and adult parasites of lungworm species that inhabit the lung parenchyma, such as *M. capillaris* [35,38,39]. Although anthelmintics with less than 100% efficacy may be useful in controlling lungworms, their use may lead to the selection of resistant parasites [40].

Gastrointestinal parasites were identified in all herds surveyed, thereby confirming its ubiquitous distribution in the Centre region of Portugal. 

In sheep, Strongyle-type eggs were the most frequently found (69.8%), followed by *Eimeria* oocysts (40.3%). A study performed in northeast of Portugal obtained a similar prevalence of infection by Strongyle parasites (85.4%, excluding *Nematodirus*) in sheep, although *Eimeria* oocysts prevalence was slightly higher (75.8%) [4] than that obtained in the present study (68.9%). However, the enormous difference between the two studies was the burden of infection. Indeed, whereas in northeast the mean count of Strongyle-type eggs reached 415.5, in the Centre region it was 206.4 EPG (both values excluding *Nematodirus* eggs). The differences were even more significant for *Eimeria* oocysts, with a mean count of 670.1 OPG recorded in the Northeast region [4] and 99.6 OPG in the present study. 

In goats, the prevalences of Strongyle-type eggs and *Eimeria* oocysts were 87.8% and 68.9%, with a burden of infection of 518.0 EPG and 399.8 OPG, respectively. However, due to the scarcity of parasitological studies in goats and the differences in the methodology among studies, it is difficult to compare results. In a study carried out in GXTBR on pooled samples, the prevalences of Trichostrongyloidea and *Eimeria* were 76.0% and 84.0% and the burdens of infection were 107.9 EPG and 2440.5 OPG, respectively [30]. In Alentejo region, located in the South of Portugal, a high *Eimeria* spp. prevalence (99.0%) and a high burden of infection (1450 and 796 OPG in young and adult goats, respectively) were observed [23].

Although the risk of nematode infection in small ruminants is continuous throughout the year, there are peaks of egg/oocyst shedding in some periods of the year [41]. Animal-related factors, such as age, physiological phase, and nutritional status [42], and extrinsic factors, such as herd management, animal density, season, and climatic conditions, among others, can affect the prevalence of infection and egg shedding [43,44]. Thus, the differences observed between studies may be related to climatic conditions of different regions, herd and animal management, sample collection (individual and pooled samples), along with other factors. 

The dynamics of small ruminants *Eimeria* spp. infection as well as OPG counts are also determined by intrinsic and extrinsic risk factors. Young animals are particularly prone to infection and shed high amounts of oocysts [23,45,46], while adult animals become more resistant, due to the establishment of an effective immune response after previous exposure [47]. Longitudinal studies revealed seasonal variations in the prevalence of *Eimeria* spp. infection. In regions with an arid and desert climate, the prevalence of infection tends to be high during summer, due to thermal stress to which animals are subjected, and in autumn due to the increase in the humidity that favor oocyst sporulation [45]. *Eimeria* shedding of oocyst is also influenced by herd size and animal density, probably due to the increased stress experienced by animals [46]. 

Thus, the higher *Eimeria* prevalence and the burden of infection observed in GXTBR may be related to the sample collection period (autumn), which favors oocyst sporulation and infection. In the Alentejo region, the high prevalence and burden of infection are probably related to sample characterization, since 46.1% of the animals surveyed in the Alentejo region were less than 5 months old [23], whereas in the present study all the animals were older than seven months. 

Eighteen different *Eimeria* species are known to infect goats [48,49] and eleven infect sheep [50]. In line with previous studies [23,51,52], *E. arloingi* and *E. ninakholyakimovae* were the most prevalent *Eimeria* species in goats, both considered as the most pathogenic [23,53]. *E. bakuensis* and *E. ovinoidalis* were the most common species identified in sheep, which, along with *E. crandallis* (not identified in our study), are considered the most pathogenic [50].

Two Cestode species, *Moniezia expansa* and *M. benedeni*, occur in small ruminants [54,55]; in this study only *M. benedini* was identified, with higher prevalence in sheep [4] and lower prevalence in goats [30] than previously reported. The prevalence of *Trichuris* spp. obtained in this study, both in sheep and goats, was lower than previously described in Portugal [4,30]. *Skrjabinema* spp., a parasite belonging to the Oxyuridae family has been found in sheep and goats [4,30,56]; in our study, characteristic eggs were only observed with low prevalence (2.0%) in goats. *Dicrocoelium* spp. was found only in sheep and in a very low prevalence (0.3%) compared with previous studies carried out in Portugal [4]. Infection prevalence seems to be influenced by seasonal, geographical, and climatic conditions [57]. Although dicrocoeliosis is traditionally associated with dry and hot climates [58], some studies suggest an increased prevalence in mountainous pastures, with low temperatures and high precipitation [59]. 

The prevalence and the burden of parasitic infection was significantly higher in goats than in sheep, as previously observed [31,60,61]. The greater susceptibility of goats to nematode infection, both gastrointestinal and pulmonary, may be related to the low ability to develop a specific protective immune response. It has been suggested that the type of grazing (in height) may have contributed to this evolutionary specificity of the immune response against nematode parasites [31,62].

In general, the burden of gastrointestinal parasitic infection in both sheep and goats (210 and 518 EPG, respectively) was not high [42,63], indicating that probably gastrointestinal parasites are not a serious health problem in these small ruminant autochthonous breeds reared in the Centre region, as has already been reported in the Churra Galega Mirandesa autochthonous sheep breed in northeast Portugal [4] and in Iberian ibex (*Capra pyrenaica*) in GXTBR in the northwestern Iberian Peninsula [30]. The low parasitic burdens presented by these animals should be seen as an opportunity for a more sustainable antiparasitic approaches based on breeding resilient and well-adapted autochthonous small ruminants and on parasitological testing to select the herds or animals that need to be dewormed, thus delaying the emergence of anthelmintic resistance [64].

Despite regular deworming (annual or biannual) of small ruminants being widely established, some anthelminthic compounds frequently used to control gastrointestinal parasitic infection are ineffective against lungworms, particularly *M. capillaris*. Considering the high prevalence and burden of Protostrongylidae infection in the Centre region of Portugal, it is urgent to determine the economic impact of the infection and its repercussions in animal health to establish appropriate therapeutic guidelines.

The implementation of integrated and sustainable strategies for the control of gastrointestinal and pulmonary parasites allows the optimization of animal anthelmintic treatment regimens. Thus, therapeutic decisions should ideally be based on the results of parasitological tests, which allow the identification of the target parasites and the herds/animals that should be dewormed to obtain the best epidemiological and/or production benefits.

## 5. Conclusions

This study identified higher lungworm prevalence in goats than in sheep. Pasture sharing and the anthelminthic compound used in the herd were identified as significant risk factors associated with Protostrongylidae infection in sheep. Production purpose and breed were identified as non-significant risk factors associated with Protostrongylidae infection in goats. *Eimeria* pathogenic species were identified in both sheep and goats. 

## Figures and Tables

**Figure 1 animals-14-01241-f001:**
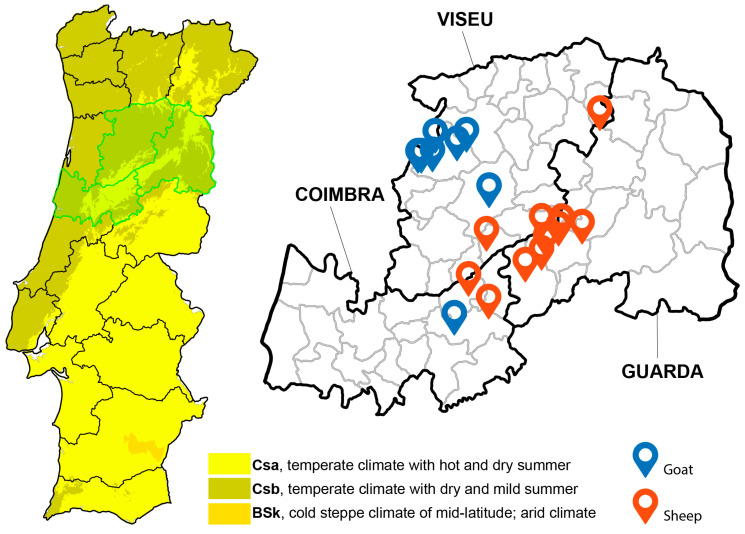
Geographical location of NUTS II Centre region of Portugal, indicating Köppen–Geiger climatic classification (1971–2000) and the specific location of sampled herds in the districts of Viseu, Guarda, and Coimbra.

**Figure 2 animals-14-01241-f002:**
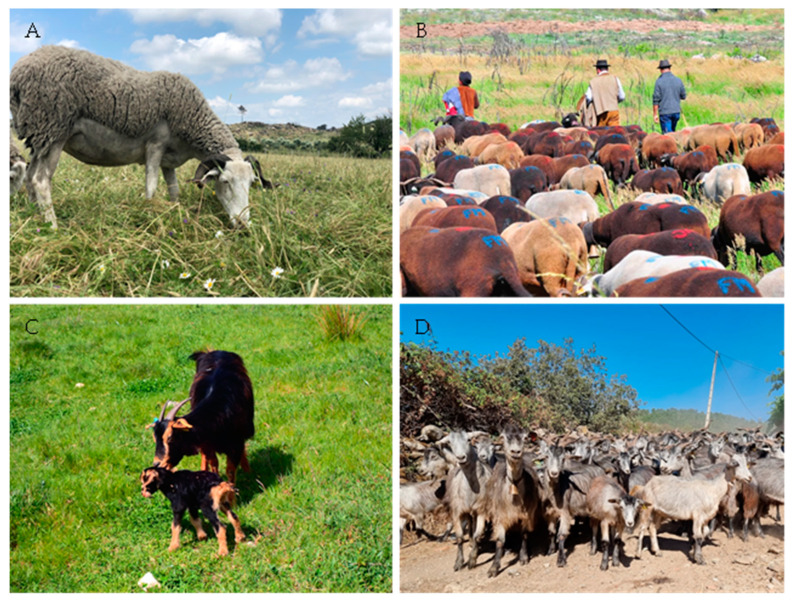
Portuguese small ruminant autochthonous breeds. (**A**)—A specimen of Serra da Estrela sheep breed. (**B**)—Transhumance of Serra da Estrela sheep breed, characterized by the seasonal movement of several herds to locations that offer better conditions during part of the year. (**C**)—A specimen of Serrana ecotype Jarmelista goat breed with her kid. (**D**)—Several specimens of Serrana ecotype Transmontano goat breed.

**Figure 3 animals-14-01241-f003:**
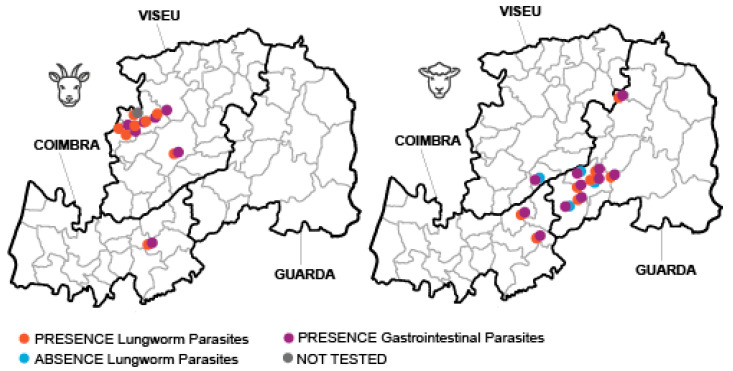
Parasitological status of sampled herds, indicating the presence/absence of lungworm and gastrointestinal parasitic infections.

**Figure 4 animals-14-01241-f004:**
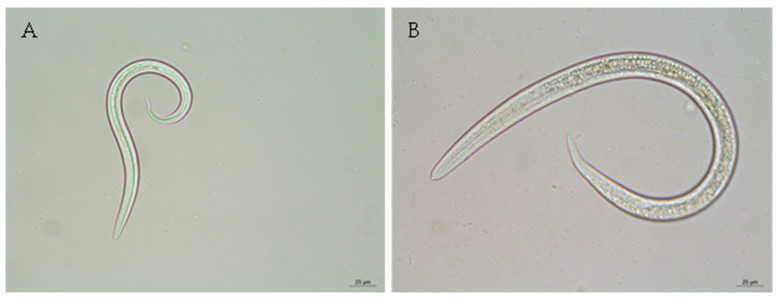
First stage larvae (L1) microphotographs of small ruminant lungworms recovered using the modified Baerman technique. (**A**)—L1 of *Muellerius capillaris*, revealing a sharply pointed tail with a spine (×400). (**B**)—L1 of *D. filaria*, showing a protuberance on the head and an unobtrusive sheath tail (×400).

**Figure 5 animals-14-01241-f005:**
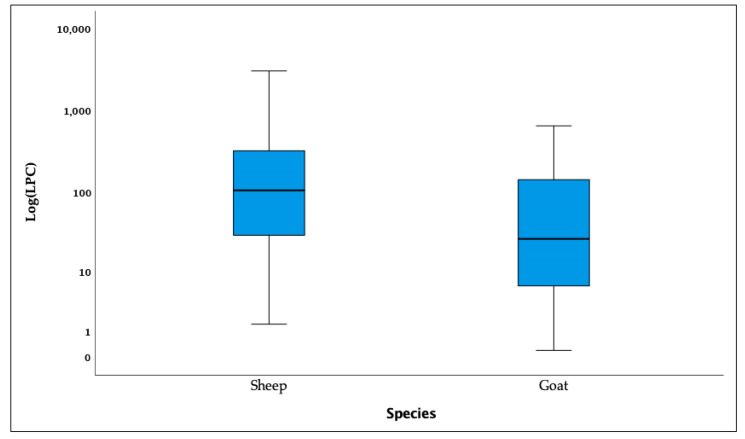
Distribution of larvae per gram of feces (LPG) by species. LPG are presented as medians, 75th and 25th percentiles, and whiskers representing the maximum and minimum values.

**Figure 6 animals-14-01241-f006:**
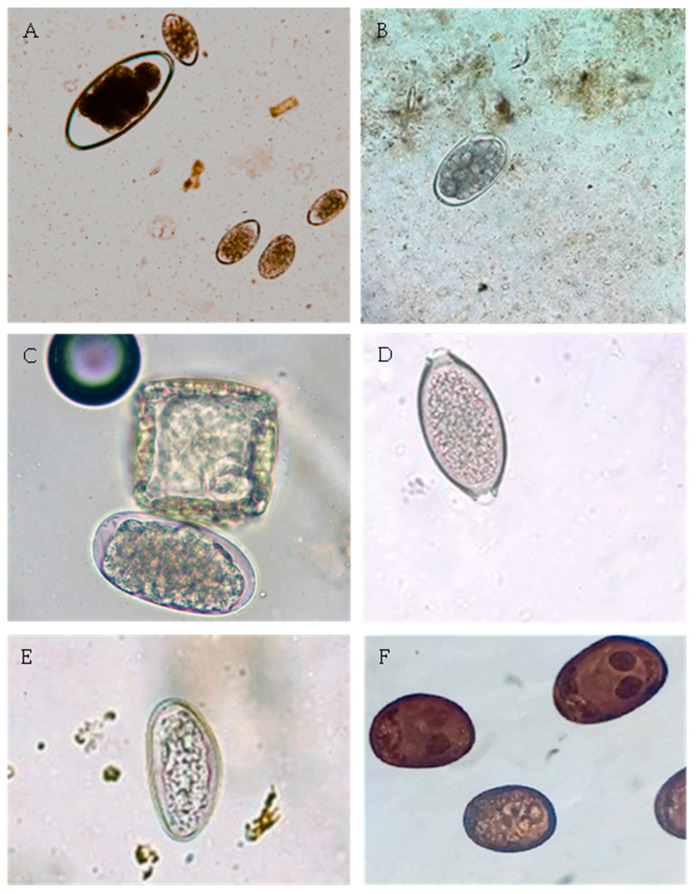
Microphotographs of small ruminant gastrointestinal parasites recovered via Mini-FLOTAC^®^. (**A**)—Strongyle-type eggs, exhibiting a characteristic *Nematodirus* spp. egg (×100). (**B**)—Strongyle-type egg (×100). (**C**)—*Moniezia benedeni* and Strongyle-type (bottom) eggs (×400). (**D**)—*Trichuris* spp. egg (×250). (**E**)—*Skrajbinema* spp. egg (×400). (**F**)—*Dicrocoelium* eggs (×250).

**Figure 7 animals-14-01241-f007:**
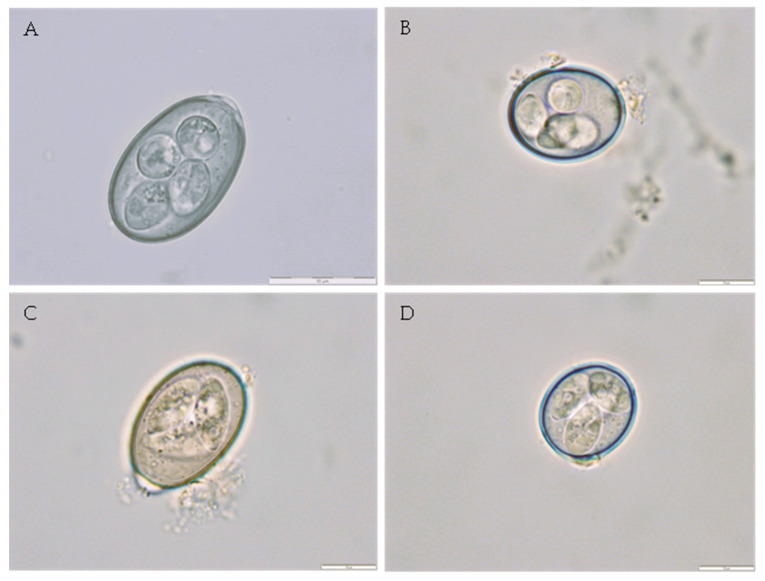
Microphotographs of *Eimeria* species recovered from small ruminant feces and *in vitro* sporulated. (**A**)—*Eimeria bakuensisa* (×1000) recovered from sheep samples. (**B**)—*Eimeria ovinoidalis* (×1000) recovered from sheep samples. (**C**)—*Eimeria arloingi* (×1000) recovered from goat samples. (**D**)—*Eimeria ninakholyakimovae* (×1000) recovered from goat samples.

**Table 1 animals-14-01241-t001:** Herd characterization, including geographical location, production system, and parasitological management.

	Sheep Herds (*n =* 14)	Goat Herds (*n =* 16)	Total Herds (*n =* 30)
District			
Viseu	0	15	15
Guarda	11	0	11
Coimbra	3	1	4
Production system			
Extensive	0	0	0
Semi-extensive	14	16	
Intensive	0	0	0
Pasture sharing			
Yes	7	0	7
No	7	16	23
Production purpose			
Milk	14	2	16
Meat	0	14	14
Animal breed			
Serra da Estrela	14	-	14
Serrana ecotype Jarmelista	-	1	1
Serrana ecotype Transmontano crossed breed	-	15	15
Parasitological testing			
Yes	1	0	1
No	13	16	29
Deworming frequency			
Twice a year	10	1	11
Annual	4	15	19
Dewormer			
Eprinomectin (Eprecis^®^)	3	1	4
Albendazole (Sinvermin^®^)	1	15	16
Mebendazole + Closantel (Seponver plus^®^)	4	0	4
Ivermectin + clorsulon (Topimec^®^; Ivomec F^®^)	6	0	6

Eprecis^®^ (Ceva Santé Animale, Loudéac, France); Sinvermin^®^ (Laboratórios Syva, S.A.U, Léon, Spain); Seponver plus^®^ (Lusomedicamenta—Sociedade Técnica Farmacêutica, S.A., Barcarena, Portugal); Topimec^®^ (Chanelle Pharmaceuticals manufacturing Ltd., Co. Galway, Irland); Ivomec F^®^ (Merial S.A.S, Toulouse, France).

**Table 2 animals-14-01241-t002:** Prevalence of lungworm infection in small ruminants from Centre region of Portugal.

Lungworm Infection	Ovine (*n =* 208)			Caprine (*n =* 203)			Total (*n =* 411)		
	n	%	CI 95%	n	%	CI 95%	n	%	CI 95%
Only *D. filaria*	3	1.4	0.4–3.8	0	-	-	7	1.7	0.8–3.3
Only Protostrongylidae	36	17.3	12.6–22.9	194	95.6	92.1–97.8	234	56.9	52.1–61.7
Mixed infection	4	1.9	0.7–4.5	0	-	-	4	0.97	0.3–2.3
Total	43	20.7	15.6–26.6	194	95.6	92.1–97.8	237	57.7	52.8–62.4

CI—Confidence interval for a proportion.

**Table 3 animals-14-01241-t003:** Risk factors associated with sheep Prostostrongylidae infection.

Variable (Category)	B	OR	Significance
Dewormer			
Eprinomectin (Eprecis^®^)		Ref.	
Albendazol (Sinvermin^®^)	3.391	29.702	0.003
Mebendazol + Closantel (Seponver plus^®^)	2.005	7.426	<0.001
Ivermectin + clorsulon (Topimec^®^; Ivomec F^®^)	2.166	8.720	<0.001
Share Pasture			
Yes		Ref.	
No	−1.704	0.182	0.001

B—Regression coefficient; OR—Odds ratio (Exp (B)). Eprecis^®^ (Ceva Santé Animale, Loudéac, France); Sinvermin^®^ (Laboratórios Syva, S.A.U, Léon, Spain); Seponver plus^®^ (Lusomedicamenta—Sociedade Técnica Farmacêutica, S.A., Barcarena, Portugal); Topimec^®^ (Chanelle Pharmaceuticals manufacturing Ltd., Co. Galway, Irland); Ivomec F^®^ (Merial S.A.S, Toulouse, France).

**Table 4 animals-14-01241-t004:** Risk factors associated with goat Prostostrongylidae infection.

Variable (Category)	B	OR	Significance
Production purpose			
Milk		Ref.	
Meat	−1.809	0.164	0.150
Breed			
Serrana ecotype Transmontano crossbreed		Ref.	
Serrana ecotype Jarmelista breed	2.234	9.333	0.011

B—Regression coefficient; OR—Odds ratio (Exp (B)).

**Table 5 animals-14-01241-t005:** Prevalence of gastrointestinal parasitic infection in small ruminants from Centre region of Portugal.

Gastrointestinal Parasitic Infection	Sheep (*n =* 159)			Goats (*n =* 148)			Total (*n =* 307)		
	*n*	%	CI	*n*	%	CI	*n*	%	CI
Strongyle type	111	69.8	62.4–76.5	130	87.8	81.9–92.4	241	78.5	73.7–82.8
*Eimeria* spp.	64	40.3	32.9–48.0	102	68.9	61.2–76.0	166	54.1	48.5–59.6
*Moniezia benedini*	6	3.8	1.60–7.6	12	8.1	4.5–13.3	18	5.9	3.6–8.9
*Trichuris*	4	2.5	0.9–5.9	1	0.7	0.1–3.1	5	1.6	0.6–3.5
*Skrjabinema*	0	-	-	3	2.0	0.6–5.3	3	1.0	0.3–2.6
*Dicrocoelium*	1	0.6	0.1–2.9	0	-	-	1	0.3	0.0–1.5

CI—Confidence interval for a proportion.

**Table 6 animals-14-01241-t006:** Burden of gastrointestinal parasitic infection in small ruminants in the Centre region of Portugal.

Burden of Infection (EPG or OPG)	Sheep (*n =* 159) (Mean ± SD)	Goats (*n =* 148) (Mean ± SD)	Total (*n =* 307) (Mean ± SD)
Strongyle type	210.4 ± 278.5	518.0 ± 564.9	376.3 ± 480.3
*Eimeria* spp.	99.6 ± 121.71	399.8 ± 586.7	284.8 ± 488.4
*Moniezia benedini*	105.8 ± 130.9	75.8 ± 53.0	85.8 ± 84.1
*Trichuris* spp.	7.5 ± 2.9	10.0	8.0 ± 2.7
*Skrjabinema* spp.	-	30.0 ± 26.5	30 ± 26.5
*Dicrocoelium dendriticum*	30	-	30

EPG—Eggs per gram of feces; OPG—Oocysts per gram of feces; SD—Standard deviation.

**Table 7 animals-14-01241-t007:** *Eimeria* species identified in sheep and goat samples after *in vitro* oocyst complete sporulation.

**Sheep**	** *n* **	**%**
*E. bakuensis*	27	67.5
*E. ovinoidalis*	18	45
*E. granulosa*	16	40
*E. parva*	14	35
*E. ahsata*	3	7.5
*E. pallida*	1	2.5
**Goats**	** *n* **	**%**
*E. arloingi*	47	90.4
*E. ninakholyakimovae*	42	80.8
*E. caprina*	22	42.3
*E. alijevi*	21	40.4
*E. christhenseni*	17	32.7
*E. caprovina*	10	19.2
*E. hirci*	9	17.3

## Data Availability

The data presented in this study are available on request from the corresponding author.

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
