# Peer review of "Pulmonary and Gastrointestinal Parasitic Infections in Small Ruminant Autochthonous Breeds from Centre Region of Portugal—A Cross Sectional Study"

_animals, 2024, doi:10.3390/ani14081241_

Round 1

Reviewer 1 Report

Comments and Suggestions for Authors

The manuscript “Pulmonary and gastrointestinal parasitic infections in small ruminant autochthonous breeds from Centre region of Portugal – a cross sectional study” presents a survey of parasite burden and type of infection across multiple goat and sheep herds sampled from three geographic districts of Portugal. This information will be of interest to small ruminant researchers, but changes are needed before this manuscript can be considered for publication.

The statistical approach of this study requires reevaluation. The main variables included in the logistic regression model are confounded by species, which may be skewing the interpretation of results. The authors report that animals dewormed with Albendazole, Mebendazole + Closantel, or Ivermectin + Clorsulon had greater risk of parasite burden, but this conclusion ignores the distribution of sheep vs. goat flocks being dewormed under each program. Sheep had lower average parasite load and were also the majority of animals dewormed with Eprinomectin (3 sheep herds vs 1 goat herd). The authors’ conclusions do not appear to be fully supported by the data; at the very least, multicollinearity must be considered or discussed. The authors identify a contradiction in their model outputs (lines 430-433) but do not adequately interrogate this issue. The current presentation is not statistically sound. 

Introduction

1.     Line 116: What is meant by the statement “the Baermann technique has a suboptimal sensitivity”? The citations referenced seem to contradict this statement [16, 17]. This should be clarified or corrected.

Materials and Methods

2.     Lines 156-160: Samples were taken at the time of deworming. For animals receiving deworming treatment twice per year, was this timepoint approx. 6 months since the last treatment? Was there any variation in time since last dewormed?

3.     Line 160: What was the range in flock/herd size? Sheep herds were >25 head, what was the maximum herd size sampled from? Were animals selected randomly from each herd? What was the range in the number of goats sampled from each herd?

4.     Line 177: regarding pasture sharing, was this sharing with other species or the same species?

5.     Line 192: is “gaze” a typo of “gauze”?

6.     Line 195: is it correct that only 5 larvae from each sample were used to identify the type of lungworm infection? Is this a standard practice that should be cited?

7.     Line 240: the phrasing here is unclear what is meant by “previously removed”, and perhaps this statement is better suited to the Results section (lines 290-297)?

8.     Lines 280-283: It would be valuable to have more descriptive information provided regarding parasite burden. Can the number of animals with light / moderate / heavy infection be reported? 

9.     Lines 293-297: My main concern with the results of this study is the number of variables that seem to be confounded by species. District, production purpose, deworming frequency, and dewormer type are nearly entirely dependent on species. Was multicollinearity between predictor variables investigated? 

10.  Table 1: can the number of animal records for each data point be added, perhaps as “number flocks (number head of animal)”? This would aid in interpretation of Table 3, where animal counts are provided for presence and absence of parasite infection.

11.  Regarding Table 3, can the percentages be given on a per-row basis versus per variable? For example, give the percent of Viseu samples with presence/absence of parasites.

Discussion

12.  Lines 377-381: can this information be given in the methods section.

13.  Line 391: “all sampled goat herds (with higher prevalence of infection compared with sheep) were from Viseu district.” Did the goat herd sampled from Coimbra have a lower prevalence compared to sheep, or is there a typo here? If the incidence between flocks was compared, can these data be added to the manuscript results to support this information provided in the Discussion?

14.  Lines 430-433: “the binary logistic regression model identified annual deworming as a significant protection factor” – this outcome of the logistic regression needs to be checked. According to the numbers provided in Table 3, 78.5% of animals dewormed twice a year had an absence of parasites, compared to 19.76% of animals dewormed once per year. These statistics do not make sense.

Comments on the Quality of English Language

Overall the English is clear and understandable. There are some minor typographical errors. 

Author Response

REVIWER 1

The manuscript “Pulmonary and gastrointestinal parasitic infections in small ruminant autochthonous breeds from Centre region of Portugal – a cross sectional study” presents a survey of parasite burden and type of infection across multiple goat and sheep herds sampled from three geographic districts of Portugal. This information will be of interest to small ruminant researchers, but changes are needed before this manuscript can be considered for publication.

The statistical approach of this study requires reevaluation. The main variables included in the logistic regression model are confounded by species, which may be skewing the interpretation of results. The authors report that animals dewormed with Albendazole, Mebendazole + Closantel, or Ivermectin + Clorsulon had greater risk of parasite burden, but this conclusion ignores the distribution of sheep vs. goat flocks being dewormed under each program. Sheep had lower average parasite load and were also the majority of animals dewormed with Eprinomectin (3 sheep herds vs 1 goat herd). The authors’ conclusions do not appear to be fully supported by the data; at the very least, multicollinearity must be considered or discussed. The authors identify a contradiction in their model outputs (lines 430-433) but do not adequately interrogate this issue. The current presentation is not statistically sound.

The authors would like to thank reviewer 1 for his/her comments, which contributed to improving the statistical analysis of the paper. To prevent the animal species from acting as a confounding factor, two logistic regression models were constructed, one for sheep and one for goats. The accuracy of the two independent models decreased compared to the previous model as suggested by reviewer 3.

Introduction

  1. Line 116: What is meant by the statement “the Baermann technique has a suboptimal sensitivity”? The citations referenced seem to contradict this statement [16, 17]. This should be clarified or corrected.

One of the citations was changed and sensitivity values of Baermann technique were included to clarify the statement (lines 119 and 120).

Materials and Methods

  1. Lines 156-160: Samples were taken at the time of deworming. For animals receiving deworming treatment twice per year, was this timepoint approx. 6 months since the last treatment? Was there any variation in time since last dewormed?

All herds included in the study have not been dewormed in the previous six months. This information was added to the text (line 169).

  1. Line 160: What was the range in flock/herd size? Sheep herds were >25 head, what was the maximum herd size sampled from? Were animals selected randomly from each herd? What was the range in the number of goats sampled from each herd?

The requested information was included in the text (lines 170-179).

  1. Line 177: regarding pasture sharing, was this sharing with other species or the same species?

Seven sheep herds shared pasture with other small ruminant herds (sheep and goats). This information was included in the text (lines 191-192).

  1. Line 192: is “gaze” a typo of “gauze”?

Thanks for noticing.

  1. Line 195: is it correct that only 5 larvae from each sample were used to identify the type of lungworm infection? Is this a standard practice that should be cited?

When we began this work, we were unaware of the species of lungworms that circulated in this region in small ruminants, so we were extremely cautious with the morphological identification. L1 identification was based on microscopic observation of all larvae counted. Furthermore, at least 5 larvae from each animal were immobilized and photographed to confirm identification. The text has been slightly edited to clarify the message (lines 217-223). Thanks for noticing.

  1. Line 240: the phrasing here is unclear what is meant by “previously removed”, and perhaps this statement is better suited to the Results section (lines 290-297)?

The sentence was removed, because it no longer makes sense in the new statistical analysis.

  1. Lines 280-283: It would be valuable to have more descriptive information provided regarding parasite burden. Can the number of animals with light / moderate / heavy infection be reported?

The burden of infection was characterized (lines 256-258 in material and methods section and in results section (317-319) and a box plot (Figure 5) was added to show the distribution of LPG value for both sheep and goats (lines 327-336).

  1. Lines 293-297: My main concern with the results of this study is the number of variables that seem to be confounded by species. District, production purpose, deworming frequency, and dewormer type are nearly entirely dependent on species. Was multicollinearity between predictor variables investigated?

To resolve the confusing factor “species”, two multivariable logistic regression models were constructed, one for sheep and one for goats as suggested by reviewer 3. Only two variables remained in each model: production purpose and breed in goats and pasture sharing and dewormer.

  1. Table 1: can the number of animal records for each data point be added, perhaps as “number flocks (number head of animal)”? This would aid in interpretation of Table 3, where animal counts are provided for presence and absence of parasite infection.

The number of animals was introduced in table 3 and 4 (first column) to aid in the interpretation of inferential statistical analysis and risk analysis.

  1. Regarding Table 3, can the percentages be given on a per-row basis versus per variable? For example, give the percent of Viseu samples with presence/absence of parasites.

The new tables created (table 3 and 4) contain the total number of animals per category, as well as the % of positives and negatives per variable and category.

Discussion

  1. Lines 377-381: can this information be given in the methods section.

The information was transferred to methods section (lines197-201).

  1. Line 391: “all sampled goat herds (with higher prevalence of infection compared with sheep) were from Viseu district.” Did the goat herd sampled from Coimbra have a lower prevalence compared to sheep, or is there a typo here? If the incidence between flocks was compared, can these data be added to the manuscript results to support this information provided in the Discussion?

The paragraph you point to, refers to the total number of animals (sheep + goats). As all goats were sampled in the district of Viseu and practically all were positive, the global prevalence was higher in this district. However, as it was necessary to redo the statistical analysis, the paragraph in question no longer made sense and was removed. Thanks!

  1. Lines 430-433: “the binary logistic regression model identified annual deworming as a significant protection factor” – this outcome of the logistic regression needs to be checked. According to the numbers provided in Table 3, 78.5% of animals dewormed twice a year had an absence of parasites, compared to 19.76% of animals dewormed once per year. These statistics do not make sense.

Removing the confounding factor (species), the new regression models constructed did not include deworming frequency as protection/risk factor. Thanks for your constructive comment.

Comments on the Quality of English Language

Overall the English is clear and understandable. There are some minor typographical errors.

The typographical errors were corrected.

Reviewer 2 Report

Comments and Suggestions for Authors

This is an interesting manuscript which adds to the limited data on the rarer breeds of animals from around the world, this time focussing on Portuguese sheep and goats. It is generally well written and well executed, with a nice number of samples included and nice analysis.

I have a few very minor comments below which are mainly linked to grammatical issues. I have tried to make a suggestion below as to what to change it it.

Line 36- breeds in the Centre region of. …. (reword)

Line 40- three districts of the Centre region ….. (reword)

Line 48- health in the Centre region of … (reword)

Line 50- breeds in the Centre region of Portugal …. (Reword)

Line 54- subjected to the modified Baermann test …. (reword)

Line 64- health in the Centre region …(reword)

Line 71- impact on the local economy … (reword)

Line 75- most of them are in danger of extinction …. (reword)

Line 84- reared for a dairy purpose …(reword)

Line 106- are coughed up and swallowed …(reword)

Line 107- excreted in feces …. (reword)

Line 116- inexpensive Baermann technique ….. (reword)

Line 122- emergence of anthelminthic resistance …(reword)

Line 130- contribute to increasing our knowledge about …(reword)

Line 136- Coimbra extend through an area ….(reword)

Line 137- Perhaps should read something like …..’The Centre region is characterised by …..’ (reword)

Line 140- the Centre region ….. (reword)

Line 140- and 140 please define what Cs, Csb and Csa are

Line 141- The small ruminant population….(reword)

Line 143- In this region, the production system is composed of small size herds …(reword)

Line 147- tradition in the Centre region …(reword)

Line 159- I think this should be ‘Herd selection was carried out for convenience, and inclusion criteria included sheep herds with 20 animals or more, and being visited by the official veterinary brigade between March and June 2023.’

Line 321, 323 and 325- maybe single sounds better here than simple?

Line 392- from the Visea district …. (reword)

Line 397- interpreted with caution may sound better here? (reword)

Line 407- is this lugworm or lungworm?

Line 408- higher than that observed in Spain …(reword)

Line 432- However, the binary logistic regression…. (reword)

Line 448- prevalence was slightly higher (75.8%) [4] than that obtained …..(reword)

Line452- including some stats here maybe nice to further prove the point?

Line 458- can you please define GXTBR (I apologise if I missed it)

Line 480- shedding of oocysts is also ….. (reword)

Line 504- studies suggest an increased prevalence …. (reword)

Comments on the Quality of English Language

These are detailed above

Author Response

REVIWER 2

This is an interesting manuscript which adds to the limited data on the rarer breeds of animals from around the world, this time focussing on Portuguese sheep and goats. It is generally well written and well executed, with a nice number of samples included and nice analysis.

Thank you very much!

I have a few very minor comments below which are mainly linked to grammatical issues. I have tried to make a suggestion below as to what to change it it.

All grammatical issues have been resolved. Thanks!

Line 36- breeds in the Centre region of. …. (reword).

Line 40- three districts of the Centre region ….. (reword)

Line 48- health in the Centre region of … (reword)

Line 50- breeds in the Centre region of Portugal …. (Reword)

Line 54- subjected to the modified Baermann test …. (reword)

Line 64- health in the Centre region …(reword)

Line 71- impact on the local economy … (reword)

Line 75- most of them are in danger of extinction …. (reword)

Line 84- reared for a dairy purpose …(reword)

Line 106- are coughed up and swallowed …(reword)

Line 107- excreted in feces …. (reword)

Line 116- inexpensive Baermann technique ….. (reword)

Line 122- emergence of anthelminthic resistance …(reword)

Line 130- contribute to increasing our knowledge about …(reword)

Line 136- Coimbra extend through an area ….(reword)

Line 137- Perhaps should read something like …..’The Centre region is characterised by …..’ (reword)

Line 140- the Centre region ….. (reword)

Line 140- and 140 please define what Cs, Csb and Csa are

The description of the Köppen-Geiger classification has been improved (lines 143-148).

Line 141- The small ruminant population….(reword)

Line 143- In this region, the production system is composed of small size herds …(reword)

Line 147- tradition in the Centre region …(reword)

Line 159- I think this should be ‘Herd selection was carried out for convenience, and inclusion criteria included sheep herds with 20 animals or more, and being visited by the official veterinary brigade between March and June 2023.’

The text was changed to include information requested by reviewer 1.

Line 321, 323 and 325- maybe single sounds better here than simple?

 Thanks for your suggestion.

Line 392- from the Visea district …. (reword)

Line 397- interpreted with caution may sound better here? (reword)

Line 407- is this lugworm or lungworm?

Line 408- higher than that observed in Spain …(reword)

Line 432- However, the binary logistic regression…. (reword)

Line 448- prevalence was slightly higher (75.8%) [4] than that obtained …..(reword)

Line452- including some stats here maybe nice to further prove the point?

Line 458- can you please define GXTBR (I apologise if I missed it)

The acronym has already been defined (line 479).

Line 480- shedding of oocysts is also ….. (reword)

Line 504- studies suggest an increased prevalence …. (reword)

Reviewer 3 Report

Comments and Suggestions for Authors

The manuscript titled "Pulmonary and gastrointestinal parasitic infections in small ruminant autochthonous breeds from Centre region of Portugal - a cross sectional study" mainly presents pulmonary nematodes prevalence and intensity of infection in sheep and goats in Centre Portugal. It is important to maintain information about these type of parasites, not commonly studied in science. So, the tha manuscript is interesting amd it should be published... after some corrections that must be applied on.

I'll indicte in order:

1. I think NUTS needs a reference, as I try to find this classification and I've found some. To avoid this confusion, please include a reference or web URL

2. I also think that Köppen-Geiger classification needs a reference; it is in the same situation than NUTS

3. In the sentence "The amount of feces collected from 148 sheep and 159 was sufficient..." I think you have lost 159 GOATS...

4. The most important thing: Statistics. 2.8. Data processing... I think that taking into account that you have a small number of independent variables, Chi squared test is not necessary before the logistic regression. I think it should be better to apply directly a logistic regression and then define the best model with a backward-forward stepwise technique. Of couse, you cannot apply the logistic regression to sheep and goats as a unique variable. The animals are different; location were different; parasites behave differently... I think that you can include in sheep prevalence analysis an independent variable taking into account the effect that goat make over sheep when the two are maintained together (Reference 31). So, you have to analyse sheep and goat separately (two logistic regression). This is the reason why district appears as a risk factor; Viseu is in risk, because in Viseu you have only goats (being a goat is a risk; take it as a joke). Consider this also in discusion. We also include GIN infection as a independent variable, as Trychostrongilidea have been indicated as immunodepresor (only as a suggestion; of course, this could be a problem with a lot of NA animals (animals without GIN analysis; out of the logistic test)). THIS IS THE MOST IMPORTANT RECOMMENDATION

5. where is reference 36? I cannot find it in text

6. I think that indicating the proportion of Muellerius among pulmonary nematodes in sheep and goats could be important. We have find that in sheep there is almost only Muellerius (GIN treatment applied from almost 30 years), while in goats another species can be found sometimes (GIN treatment began years after).

7. Conclusions. Conclusions do not cover Objectives. You have indicated clearly 3 objectives (1. Prevalence and burden of lungworm infection; 2. Risk factors 3. prevalence and burden of GIN, all of then in samll ruminants). Conclusions cover onlypart of the first one. I think you need to include in conclusions data as answer of these three  objectives.

I think that manuscripts as this are very important for clinicians and farmers, but I think you must improve statistics, risk analyses and conclusions.

Best regards,

Author Response

REVIWER 3

The manuscript titled "Pulmonary and gastrointestinal parasitic infections in small ruminant autochthonous breeds from Centre region of Portugal - a cross sectional study" mainly presents pulmonary nematodes prevalence and intensity of infection in sheep and goats in Centre Portugal. It is important to maintain information about these types of parasites, not commonly studied in science. So, the manuscript is interesting, and it should be published... after some corrections that must be applied on.

Thanks. In fact, parasitic infections are poor studied in these animal species, particularly in goats. In our region there were no data available.

I'll indicte in order:

  1. I think NUTS needs a reference, as I try to find this classification and I've found some. To avoid this confusion, please include a reference or web URL

A reference was included (line 71).

  1. I also think that Köppen-Geiger classification needs a reference; it is in the same situation than NUTS

There is already a reference in the text (now ref. 26) (line 148).

  1. In the sentence "The amount of feces collected from 148 sheep and 159 was sufficient..." I think you have lost 159 GOATS...

Thanks for noticing!

  1. The most important thing: Statistics. 2.8. Data processing... I think that taking into account that you have a small number of independent variables, Chi squared test is not necessary before the logistic regression. I think it should be better to apply directly a logistic regression and then define the best model with a backward-forward stepwise technique. Of couse, you cannot apply the logistic regression to sheep and goats as a unique variable. The animals are different; location were different; parasites behave differently... I think that you can include in sheep prevalence analysis an independent variable taking into account the effect that goat make over sheep when the two are maintained together (Reference 31). So, you have to analyse sheep and goat separately (two logistic regression). This is the reason why district appears as a risk factor; Viseu is in risk, because in Viseu you have only goats (being a goat is a risk; take it as a joke). Consider this also in discusion. We also include GIN infection as a independent variable, as Trychostrongilidea have been indicated as immunodepresor (only as a suggestion; of course, this could be a problem with a lot of NA animals (animals without GIN analysis; out of the logistic test)). THIS IS THE MOST IMPORTANT RECOMMENDATION

Thank you very much for your recommendation. Two models were created (one for sheep and one for goats).

Infection with gastrointestinal parasites was identified as a risk factor (OR=1.89) in a recent study (García-Dios et al, 2021), but in our study it was not possible to include this variable as an independent variable because not all animals were tested for gastrointestinal parasites.

  1. where is reference 36? I cannot find it in text.

“Thank for noticing. Ref 36 corresponds to the text: “Pulmonary pathological changes induced by M. capillaris, characterized by the formation of inflammatory nodules may protect parasites, preventing anthelmintic therapeutic concentrations at lung parenchyma [36]”.

  1. I think that indicating the proportion of Muellerius among pulmonary nematodes in sheep and goats could be important. We have find that in sheep there is almost only Muellerius (GIN treatment applied from almost 30 years), while in goats another species can be found sometimes (GIN treatment began years after).

In this geographical region the scenario was slightly different. In sheep we identified predominantly Muellerius, Cystocaulus in one animal and Dictyocaulus in some animals from a flock. Only Muellerius was identified in goats. Unfortunately, we were unable to describe the history of infection by lungworms because this is the first study carried out in the region and I believe the second carried out in Portugal, nor relate the results with the evolution of deworming as the Spanish colleagues did so well in Galicia.

The information requested was included in the text (lines 310, 311).

  1. Conclusions. Conclusions do not cover Objectives. You have indicated clearly 3 objectives (1. Prevalence and burden of lungworm infection; 2. Risk factors 3. prevalence and burden of GIN, all of then in samll ruminants). Conclusions cover onlypart of the first one. I think you need to include in conclusions data as answer of these three objectives.

The conclusion was reformulated to meet the objectives (619-624).

I think that manuscripts as this are very important for clinicians and farmers, but I think you must improve statistics, risk analyses and conclusions.

Thank you very much. Statistics were improved as well as conclusions.

Round 2

Reviewer 1 Report

Comments and Suggestions for Authors

The authors have thoroughly addressed all of my concerns. The changes implemented greatly improve the soundness of this manuscript, it is now acceptable for publication. 

Author Response

Thank you

Reviewer 3 Report

Comments and Suggestions for Authors

Dear authors,

you have modified most of the "defects" of the manuscript, but I think you need to change two things more.

1. Although you say you have applied a logistic regressión, table 3 and 4 presents Chi squared test results. These tables show different results than what you say in text, so you have to change table 3 and 4 for the tables obtained from final logistic models. I include an old one I have when I used SPSS as an example (more than a decade, so table aspect could be different, but results should be the same, as mathematic is eternal. Nowadays I use R with much more algorithms). I suppose you have obtained similar tables for sheep and goats. If not, tables and text indicate different results and the manuscript is difficult to be understood. (In the example I send you can see that SPSS treated Estimate as B, so OR, calculated as exponetial of Estimate, is ExtB)

2. Conclusions

Again, objectives are clearly 1. Prevalence and burden of lungworm infection; 2. Risk factors 3. Prevalence and burden of GIN. Conclusions AGAIN cover only part of the first and the second. I recomend something like: Lungworm prevalence is higher in sheep than in goats. Prevalence and burden of Protostrongylidae infection in sheep are influenced by.... and in goats by .... The prevalence and the burden of parasitic infection was significantly higher in goats than in sheep, but the infection is under healthy level. If you consider interesting, you could also include something about Eimeria, above all if you consider that you find some pathogenic species (but, in that case, include it also in results).

Please consider only this two things; tables 3 and 4 adequate for the text you include in results (and related to the final logistic model in sheep and goats) and Conclusions covering objectives.

Best regards

Author Response

Dear authors,

you have modified most of the "defects" of the manuscript, but I think you need to change two things more.

The authors would like to thank the reviewer for their attentive and experienced review.

  1. Although you say you have applied a logistic regressión, table 3 and 4 presents Chi squared test results. These tables show different results than what you say in text, so you have to change table 3 and 4 for the tables obtained from final logistic models. I include an old one I have when I used SPSS as an example (more than a decade, so table aspect could be different, but results should be the same, as mathematic is eternal. Nowadays I use R with much more algorithms). I suppose you have obtained similar tables for sheep and goats. If not, tables and text indicate different results and the manuscript is difficult to be understood. (In the example I send you can see that SPSS treated Estimate as B, so OR, calculated as exponetial of Estimate, is ExtB)

The authors understand the reviewer's point of view. Therefore, the original tables 3 and 4 were placed as supplementary tables (Table S1 and Table S2, respectively) and two new tables were created (Table 3 and Table 4) only with the regression results. By the way, we have not received the example you sent us. We hope you enjoy our tables.

  1. Conclusions

Again, objectives are clearly 1. Prevalence and burden of lungworm infection; 2. Risk factors 3. Prevalence and burden of GIN. Conclusions AGAIN cover only part of the first and the second. I recomend something like: Lungworm prevalence is higher in sheep than in goats. Prevalence and burden of Protostrongylidae infection in sheep are influenced by.... and in goats by .... The prevalence and the burden of parasitic infection was significantly higher in goats than in sheep, but the infection is under healthy level. If you consider interesting, you could also include something about Eimeria, above all if you consider that you find some pathogenic species (but, in that case, include it also in results).

The conclusions were written as suggested, and a sentence was added about Eimeria infection. The most prevalent species were included in the results (lines 442 and 446) and the discussion states that the most pathogenic were identified in our samples.

Although producers on the farms we visited did not report clinical signs suggestive of lungworm infection, the truth is that producers and veterinarians recognize the positive effect of deworming on increasing production and the body condition of animals, so we do not have data that allow us to state that the infection seems to be under a healthy level.

Please consider only this two things; tables 3 and 4 adequate for the text you include in results (and related to the final logistic model in sheep and goats) and Conclusions covering objectives.

Once again, we thank reviewer 3 for his comments and suggestions.

Best regards